EMBO
Molecular Medicine

# Minimally-invasive implantable device enhances brain cancer suppression

Xiaona Cao[1,2], Jie Li[1,2], Jinliang Ren[1,2], Jiajin Peng[2], Ruyue Zhong[2], Jiahao He[1,2], Ting Xu[1,2], Zhenhua Yu[3], Huawei Jin[3], Siqi Hao[4], Ruiwei Liu[4] & Bingzhe Xu[1,2]✉

## Abstract

**Current brain tumor treatments are limited by the skull and BBB, leading to poor prognosis and short survival for glioma patients. We introduce a novel minimally-invasive brain tumor suppression (MIBTS) device combining personalized intracranial electric field therapy with in-situ chemotherapeutic coating. The core of our MIBTS technique is a wireless-ultrasound-powered, chip-sized, lightweight device with all functional circuits encapsulated in a small but efficient "Swiss-roll" structure, guaranteeing enhanced energy conversion while requiring tiny implantation windows ( ~ 3 × 5 mm), which favors broad consumers acceptance and easy-to-use of the device. Compared with existing technologies, competitive advantages in terms of tumor suppressive efficacy and therapeutic resolution were noticed, with maximum ~80% higher suppression effect than first-line chemotherapy and 50–70% higher than the most advanced tumor treating field technology. In addition, patient-personalized therapy strategies could be tuned from the MIBTS without increasing size or adding circuits on the integrated chip, ensuring the optimal therapeutic effect and avoid tumor resistance. These groundbreaking achievements of MIBTS offer new hope for controlling tumor recurrence and extending patient survival.**

**Keywords** Intracranial Brain Tumor Inhibition; Tumor Treating Field; Customized Treatment; Brain Tumors

**Subject Categories** Biotechnology & Synthetic Biology; Cancer; Methods & Resources

## Introduction

There is no effective therapeutic treatment for invasive brain cancers and patients' prognosis remains poor after standard-of-care treatment (Hausmann et al, 2023; Lin et al, 2022; Touat et al, 2020; Venteicher et al, 2017; Yanchus et al, 2022). The clinical therapy strategies for brain tumors mainly rely on surgical resection, combined with chemotherapy and radiotherapy (Chakroun et al, 2018), however, the benefits of these therapies are controversial and limited. Given the infiltrating and diffuse biology of high-grade tumors, no surgery can completely remove all tumor cells from the patient's brain and high resistance to chemotherapy/radiotherapy (Ji et al, 2023; K. Yang et al, 2022), resulting in almost 100% recurrence of glioblastoma (Adib, Ebner, Bornemann, Hempel, and Tatagiba, 2019; Faustino, Viani, and Hamamura, 2020; Gilbert et al, 2013). Due to the high recurrence rate and the lack of effective treatment, most patients diagnosed with glioblastoma survive less than 2 years and the 5-year survival rate is less than 10% (Bondy et al, 2008; Guo et al, 2022; Ostrom et al, 2017; Wu et al, 2021). More-efficient novel brain tumor suppression technologies that can significantly reduce mortality is urgently demanded.

In recent years, tumor treating field (TTFs) has received increasing attention as a novel and effective strategy for the treatment of brain tumors (Rominiyi et al, 2021; Stupp et al, 2012; Suh et al, 2020; Y. Yang et al, 2022), which employs alternating electric fields to selectively interrupt the division of cancer cells and presents positive therapeutic effect even on the most aggressive malignant brain tumors (Hinchet et al, 2019a; Stupp et al, 2015; Taphoorn et al, 2018). For nearly a decade, TTF was the only therapy proven to prolong patient survival, making it a promising technology expected to revolutionize brain tumor treatment(Kirson, Dbalý, et al, 2007; Sprugnoli et al, 2019; Stupp et al, 2017). However, TTF therapy still cannot completely replace traditional cancer treatment modalities. Therefore, an increasing number of studies explore TTF combined with chemotherapy to further improve the therapeutic effect. Chemotherapy is one of the most commonly used tumor treatment methods in clinical practice, but it exhibits obvious limitations, such as side effects and poor therapeutic effect. It is reported that the combined use of TTF and chemotherapy can reduce the side effects of chemotherapy on patients and significantly prolong the survival rate of patients (Stupp et al, 2017).

Since traditional TTF technologies still suffer from difficulties in low efficiency, rough spatial resolution and treatment inconvenience, several innovative brain tumor treatment techniques have received considerable attentions, one of which is the

[1]School of Biomedical Engineering, Sun Yat-sen University, No. 135, Xingang Xi Road, Guangzhou 510275, P.R. China. [2]School of Biomedical Engineering, Shenzhen Campus of Sun Yat-sen University, No.66, Gongchang Road, Guangming District, Shenzhen 518107, P.R. China. [3]Department of Neurosurgery, The First Affiliated Hospital of Sun Yat-sen University, Guangzhou 510080, P.R. China. [4]School of Naval Architecture & Ocean Engineering, Guangzhou Maritime University, 101 Hongshan 3rd Road, Huangpu District, Guangzhou, Guangdong 510725, P.R. China. ✉E-mail: xubzh5@mail.sysu.edu.cn

ultrasound-driven implantable tumor treatment device (UP-TTD) (Y. Yang et al, 2022). The implantable UP-TTD treatment devices have initially shown great therapeutic effects with encouraging clinical research value and broad application prospects in the field of tumor treatment. However, the mode of implantation and its possibility to be combined with other treatment modalities still needs to be further explored. Aiming to the dilemma of current brain tumor treating strategies, this study proposes a new generation of implantable ultrasound-driven tumor therapy device, which combines intracranial tumor electrical therapy with in situ implanted chemotherapy to achieve more effective inhibition of tumor cells. The novel MIBTS technology not only helps to expand the upper limit of the implantation range of the current UP-TTD technology, but also facilitates the further development of implantable multi-technology combination therapies.

## Results and discussion

### MIBTS design

The MIBTS technology is designed as an implantable medical device combining ultrasound-powered tumor treating field (UP-TTF) with intracranial in situ forming chemotherapeutic coating (ISFCC), which is functionally featured by minimally invasive implantation and multimodal therapy (Fig. 1A). The basic structure of the device is a Swiss-roll cylinder stacked with several layers of functional thin films (Fig. 1B), including a wireless ultrasonic energy converter, a current collector, insulating layers and in situ forming chemotherapeutic layers. The MIBTS converts safely penetrating ultrasound energy into a tunable alternating electric field to disrupt tumor cell mitosis and achieve tumor suppression

without affecting normal brain neurons (Y. Yang et al, 2022). ISFCC technology, as an ideal controlled-release formulation, is used to enable localized and sustained drug delivery over periods of months (Fig. 1C) without systemic toxicity. The entire system was dramatically miniaturized for minimally invasive operation with all components stacked in a ~0.5–2 mm Swiss-roll structure (Fig. 2A,B). The Swiss-roll structured wireless ultrasonic transducer is the core component of MIBTS, which realizes in vivo energy conversion from ultrasound to electrical field with a 100–500 μm micro-gap (Fig. 2C). Driven by the ultrasound waves, irregular micro-scale displacements are excited on the curved polymer film and electrically charged the electrodes after their separation (Fig. 2D). The converted electric energy is collected and exported through the electric collector. Compared with the planar-structured UP-TTD, the Swiss-roll structured MIBTS device exhibited significantly enhanced output (Fig. 2E) and required a much smaller surgical window for implantation (Fig. 2F,G). We found that MIBTS with a higher number of turns yield higher output voltages (Appendix Fig. S1a), which is due to the increased presence of interlayer interactions and multiple effects within the ultrasonic field. Temperature variations of devices with different turns were also tested after continuous 8-hour ultrasonic stimulation (Appendix Fig. S1b). Although the number of turns does not significantly increase the heat of the device, an excessive number of turns would definitely enlarge the bottom diameter of the device (Appendix Fig. S1c), which would complicate implant surgery and potentially increase patient trauma. Therefore, the appropriate number of turns should be tailored according to specific requirements for practical surgical operations.

As shown in Fig. 2F, the minimum size of the general operation window requires about 3 × 5 mm for successful surgery implantation of a Swiss-roll MIBTS, ensuring minimal patient trauma and

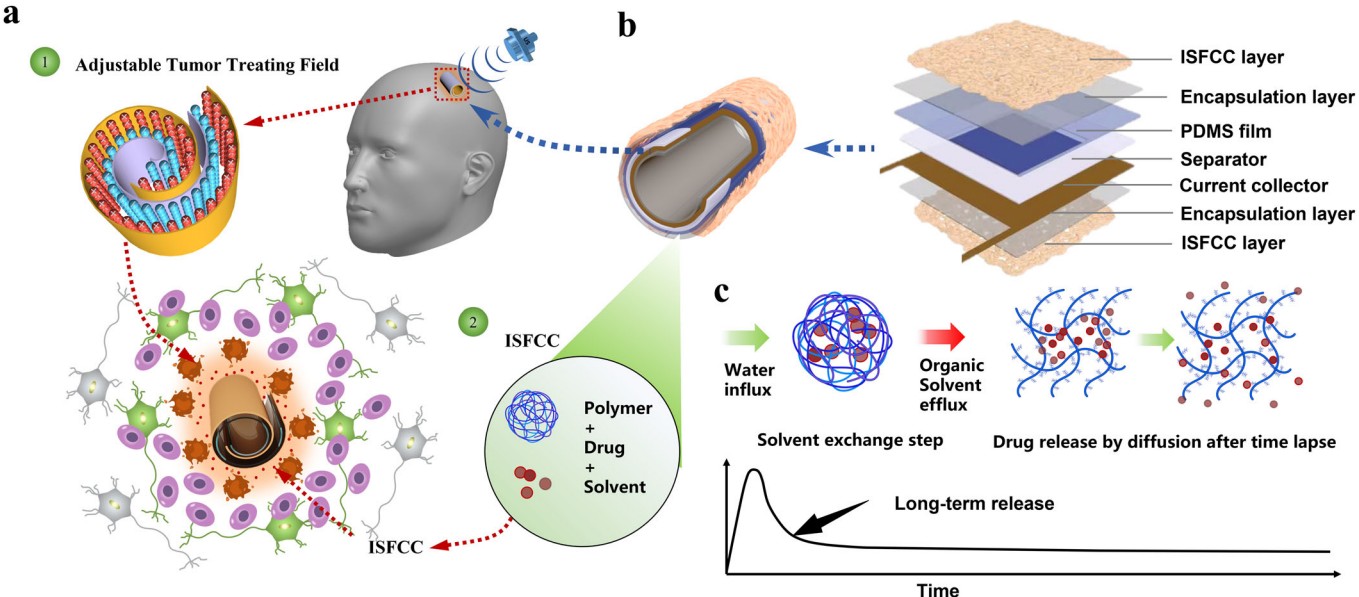

**Figure 1. Schematic and design of the flexible MIBTS.**

(A) The MIBTS technology is designed as an implantable medical device that combines ultrasound-powered tumor treating field (UP-TTF) with intracranial in situ forming chemotherapeutic coating (ISFCC). (B) The basic structure of the device is a Swiss-roll cylinder stacked with several layers of functional thin films. (C) Design of ISFCC technology to enable localized and sustained drug delivery over periods of months.

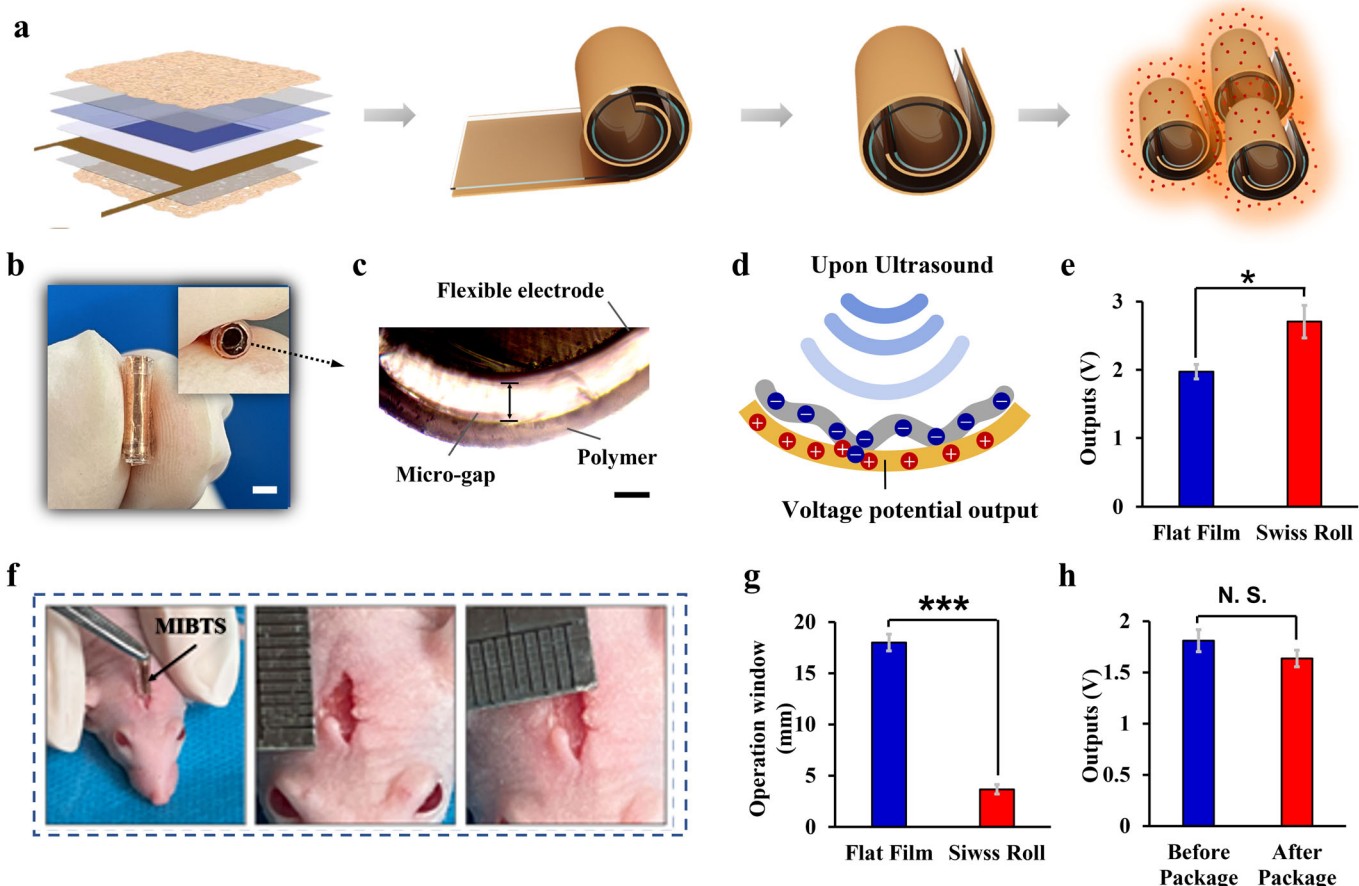

**Figure 2. Fabrication and characterization of MIBTS device.**

(A) Schematic illustration of the fabrication process. (B) The optical images of MIBTS device showing the Swiss-roll structure. Scale bar: 0.5 mm. (C) Micro-gap between the flexible electrode and polymer. Scale bar:100 um. (D) Schematic of wireless ultrasonic energy conversion. (E) The Swiss-roll structured device exhibits significantly enhanced output compared to flat film. *p value = 0.02, t-test, n = 3, error bars represent the standard deviation from three devices. (F) Characterization of general surgical window size for implantation of MIBTS. (G) The surgical window size of the Swiss-roll structured device requires a much smaller operation window. ***p value = 0.0005, t-test, n = 3, error bars represent the standard deviation from three devices. (H) A slight output decrease is observed after the PDMS package. Devices were tested at a distance of 30 mm from the ultrasound probe (frequency: 40 kHz, power density: 0.2 w/cm²). The data were analyzed using a t-test, n = 3, error bars represent the standard deviation values from three devices. Source data are available online for this figure.

shorter recovery times. For biological safety considerations, the entire device was packed with ultrathin biocompatible layers, which requires careful handling to ensure that the output efficiency of the devices was not compromised (Fig. 2H). Another function of the cladding is to protect the core components of the device from degradation and stabilize the output in complex brain environments.

## Wireless tuning of tumor-treating fields

The output intensity and frequency of MIBTS are two key parameters for effective tumor treatment. As shown in Fig. 3A, the output intensity of the Swiss-roll device reached ~4 V/cm under a 0.3 w/cm² ultrasonic excitation at 10 mm distance, which were further tuned by changing excitation distance and ultrasound frequency (Fig. 3B; Appendix Fig. S2). It can be seen that when the distance is expanded to 30 cm, attenuation still follows the inverse square law of physics (Appendix Fig. S3). Although increasing field

intensity results in a stronger inhibition on tumor, it also brings more potential safety hazards. Therefore, tumor treatment fields of 1–3 V/cm are usually the optimal choice for most clinical treatments(Wang, Pandey, and Ballo, 2019). Here, we successfully demonstrated tuning field intensities from ~1.4 V/cm to ~5.2 V/cm (Fig. 3C) by altering excitation ultrasound parameters (refer to our previous intensity tuning technique (Y. Yang et al, 2022)). The frequency of electric fields also has a decisive impact on the therapy efficiency. Low-frequency electric fields (<1 kHz) disrupt membrane polarization and destruct neuronal functions in the brain (Kirson, Dbaly, et al, 2007; Kirson et al, 2004), while high-frequency electrical fields (>1000 kHz) induce deleterious heating effects by violently vibrating intracellular polar molecules (Storm, Morton, Kaiser, Harrison, and Haskell, 1982). Here we demonstrated a tuned frequency from 100–300 kHz by adjusting the micro-gap between the flexible membrane and electrodes (Fig. 3D), which does not lead to thermal hazards nor trigger abnormal neural damages (Y. Yang et al, 2022). We control the micro-gap distance

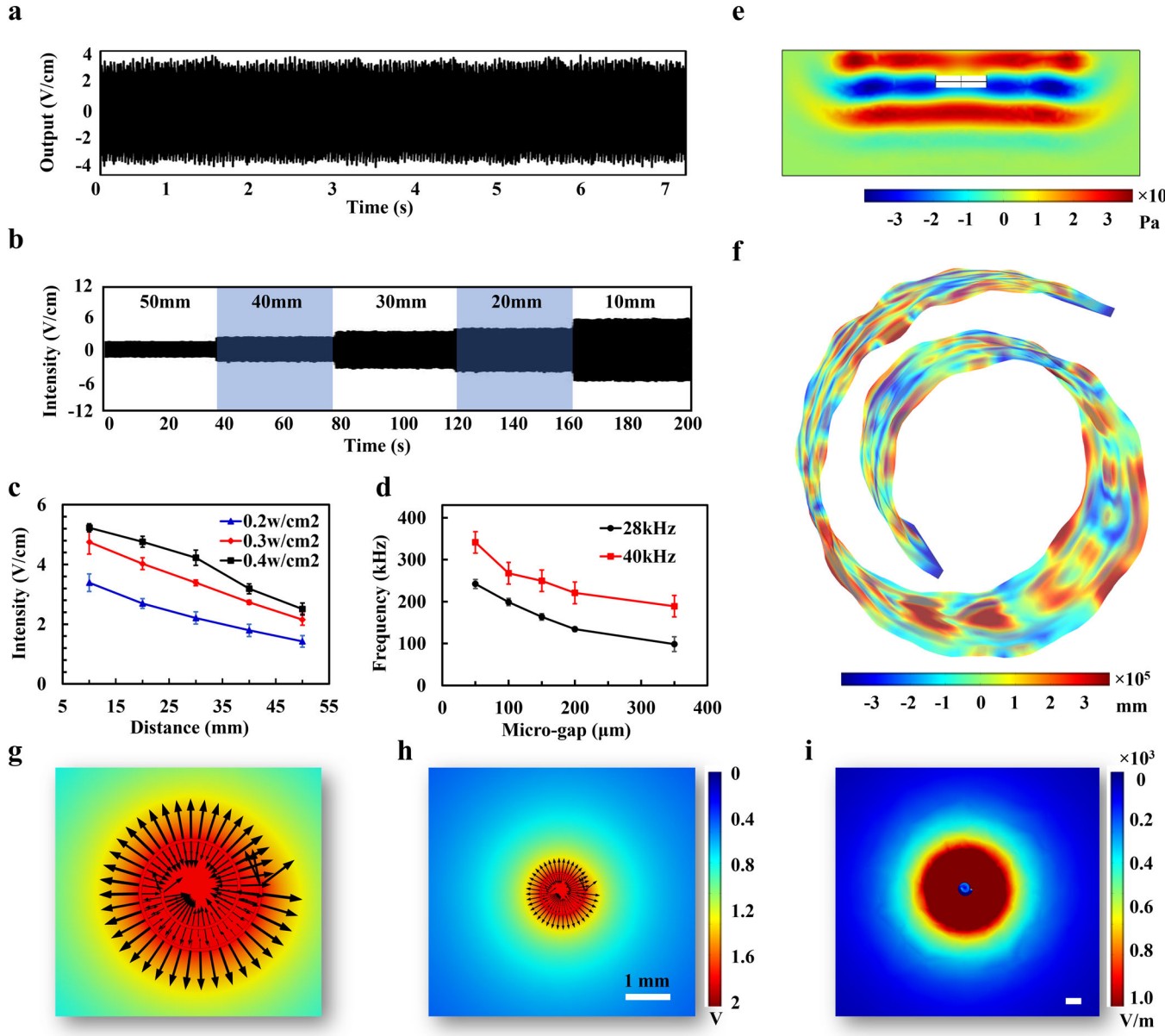

**Figure 3. Wireless tuning of tumor treating fields.**

(A) Default outputs of the device with a 0.3 w/cm² ultrasonic excitation at 10 mm. (B) The output intensity increases as the excitation distance decreases. (C) The output intensity is tuned by the exciting distance and excited ultrasonic power. The measure of center for the error bar is the mean output intensity of five devices. Error bars represent the standard deviation from the mean intensity of five devices. (D) The output frequency is tuned by the structural gap between the flexible membrane and the electrode. Error bars represent the standard deviation from three devices. (E) Simulation of the generated acoustic pressure field with an MIBTS device. (F) The simulated results on the displacement of the Swill-roll structured membrane. (G–I) The electrical field generated by the device is presented as a circular field centered on the device. Scale bar: 1 mm and 1 cm. Source data are available online for this figure.

through a hollow semi-rigid frame structure, which anchors the membrane on the sides of the device and is decisive for tuning the output frequency.

To further elucidate the device's behavioral response after ultrasound excitation, we numerically simulated the process and solved partial differential equations in three space variables. The whole system was subdivided into smaller and simpler parts by constructing a mesh of the objects, thus the problem is considered as numerical domains for solution with a finite number of points.

The ultrasonic acoustic pressure in the field (Fig. 3E) and the displacement of the excited polymer (Fig. 3F; Movie EV1) indicating multimode effects on the polymer film, resulting in complex multiple striking between the contact pairs. For each strike, negative charges and holes are generated with a positive voltage pulse. After separation, a reverse pulse is generated as the electrodes move away from the negatively charged polymer. To define the effective range of the device after implantation, the attenuation effect in the brain tissue is considered in Fig. 3G–I,

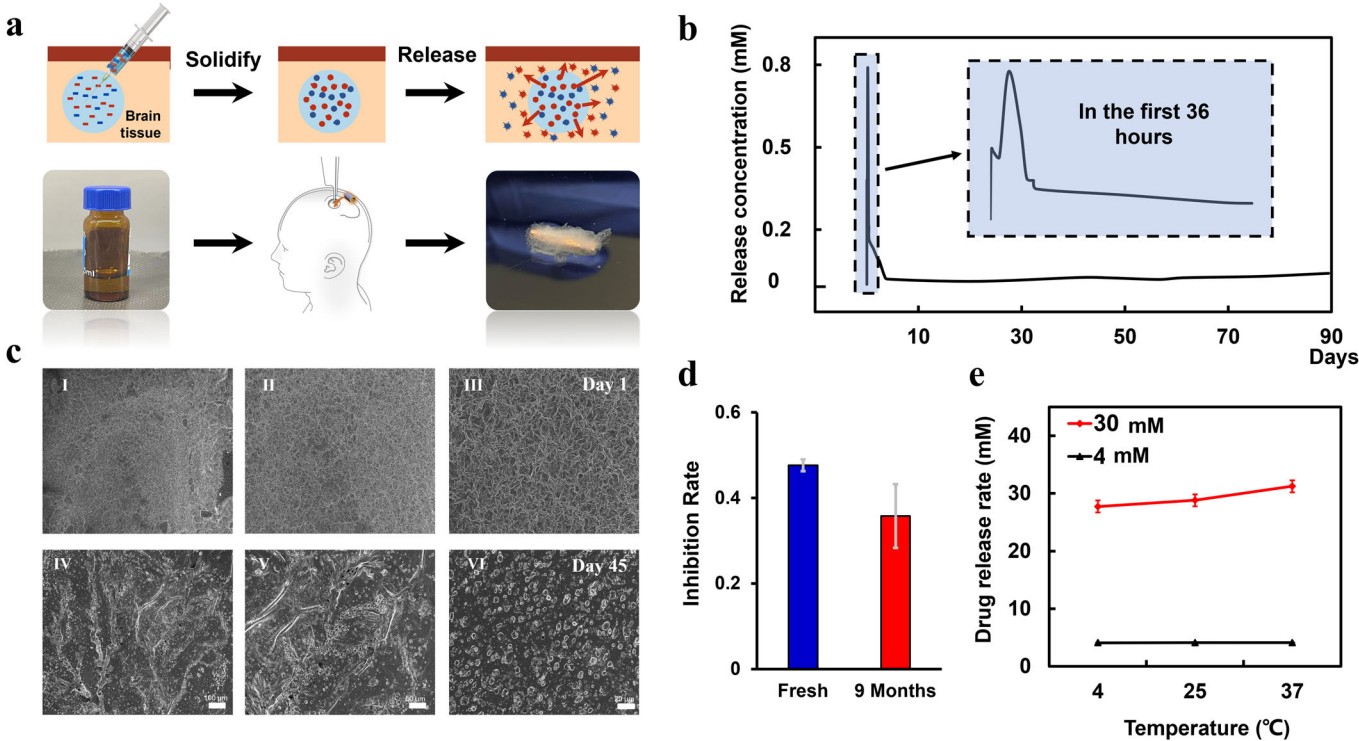

**Figure 4. Intracranial in situ forming chemotherapeutic coating.**

(A) The synthesis process and photo of the ISFC implant. (B) TMZ releases profiles over 90 days. (C) Scanning electron microscopy of ISFCC on Day 1 and Day 45. (D) ISFCC reagent exhibited long-term storage stability, with ~20% reduction in tumor inhibition after 9 months compared to freshly formulated ISFCC reagent. Error bars represent the SD from three devices. $n = 3$. (E) The release rate of ISFCC was relatively stable over the temperature range of 4–37 °C. Error bars represent the SD from three devices. $n = 3$. Source data are available online for this figure.

demonstrating a maximum working distance of up to 5.23 cm. We also simulated the electric field distribution after MIBTS was implanted in the body, and its distribution of the horizontal axis is shown in Appendix Fig. S4, suggesting that the tumor-suppressing effect is lost after the electric field attenuates by about 4 cm. Our simulation results showed an excellent agreement with the experimental data.

## Intracranial in situ forming chemotherapeutic coating

This research combines an ultrasound-powered tumor-treating device with intracranial in situ forming chemotherapeutic (ISFC) coating to effectively suppress brain tumor recurrence. An ultralong-acting tunable and biodegradable delivery coating for the UP-TTD device were constructed, offering the ability to integrate multiple drugs and sustained release delivery for months. The delivery system is established on biocompatible water-soluble organic solvents that co-dissolve biodegradable polymers and drugs. When the ISFCC is implanted into the tumor site, the ISFCC components come into contact with the liquid in the tissue, triggering a series of phase change processes (Fig. 4A). The water-soluble solvent NMP will be rapidly absorbed or diluted by body fluids, causing the concentration of the biocompatible polymer PLGA to increase and precipitate, thereby forming a solid around the tumor. During the solid formation process, drugs will be released rapidly due to the instable polymer network, forming a

burst release stage to ensure the drug takes effect quickly; Then, over time, the polymer network gradually stabilizes to form a stronger structure, and the drug molecules need to travel through a tighter polymer network to diffuse out. During this process, the drug is diffusely released from the solid implant at a steady and sustained rate until the solid implant is completely degraded after several months. This process enables sustained-release drug delivery that lasts for several months, maintaining long-term effective treatment.

The TMZ release prolife over 90 days indicates a burst release during the first 12 h followed by steady and long-term intracranial chemotherapy (Fig. 4B). The burst release is determined by the ratio of immediate-release solvents and can be adjusted as needed. We compared the release profiles of different drugs. Obviously, AL-coated ISFCs enters the stage of burst and continuous release earlier than their TMZ-coated counterparts (Appendix Fig. S5). Numerous factors play a crucial role in the drug release process. It has been reported that the burst release of drugs depends largely on in situ phase separation, which can be attributed to the physical and chemical properties of the drug, such as diffusion rate, distribution coefficient, dissociation constant, solubility, etc. (Graham, Brodbeck, and McHugh, 1999; Higuchi, 1962). in addition, it is also related to the hydrophobicity of the solvent, polymer type, molecular weight, etc. (Ma and McHugh, 2010; Patel, Carlson, Solorio, and Exner, 2010). After the burst release phase, the polymer network gradually stabilizes and forms a more robust

structure and drug molecules need to traverse a tighter polymer network to diffuse out. During this process, the drug diffuses and is released at a stable and sustained rate. At this stage, the main factors affecting drug release include drug diffusion rate within the polymer matrix, drug solubility, drug loading, matrix size (i.e., surface area of the solid implant), as well as its porosity and the tortuosity of diffusion pathways (Faisant, Siepmann, and Benoit, 2002; Higuchi, 1962).

We delved into the impact of ultrasound stimulation on drug release kinetics, subjecting AL-coated MIBTS to 12-h daily ultrasound sessions (Appendix Fig. S6). Spontaneous and stable drug release could be identified as ISFCC coating degrades, therefore, the drug release from ISFCs is a spontaneous process that can occur even without excitation. We also found that applying drugs onto coiled devices and subjecting them to ultrasound excitation would enhance their release rate. Within the initial 5 h, both groups experience an initial burst release followed by a gradual diffusion phase. Remarkably, the ultrasound group exhibits an earlier onset of both burst release and diffusion phases compared to the control, entering the degradation phase ~126 h later. Furthermore, throughout the release process, the ultrasound group consistently maintains higher drug concentrations than the control, indicative of an augmented release rate of AL-coated ISFC under ultrasound excitation.

We studied the microstructure of the ISFC coating during release using scanning electron microscopy, which revealed the slowly degraded process of the ISFC coating over time (Fig. 4C). At the same time, the preservation stability of pre-implantation preparations is also tested at three different storage conditions (5 ± 2 °C, 25 ± 2 °C, 40 ± 2 °C), including the change in physical appearance, drug concentration, activity. The ISFC formula solution were stored stable for more than 9 months without observed changes in physical appearance (Appendix Fig. S7). Meanwhile, the ISFC reagent demonstrated a strong tumor inhibitory effect in vitro even after 9 months of storage (with ~20% reduction of inhibition rate compared to the newly formulated ISFCC reagent, Fig. 4D). We also investigated the effect of temperature on the drug release rate on the MIBTS device, which indicated a stable release rate of ISFCC over the temperature range of 4–37 °C (Fig. 4E).

## Patient-personalized MIBTS therapy

We established a patient-tailored TTF and chemotherapy response platform to allow high-throughput exploration of TTF resistance and drug resistance in patients (Fig. 5A). This selection of customized TTF and chemotherapy regimens according to the patient's situation not only helps to improve treatment efficiency, but also avoids the rapid acquisition of tumor resistance. The platform was designed as a gradient microfluidic device that incorporates arrays of cylindrical processing chambers (Fig. 5B) and connected to a homemade voltage regulation control unit (Fig. 5C; Appendix Fig. S8). The construction of gradient voltage profiles (Fig. 5D,E) was realized and accordingly allows quantitative investigation of the steepness-dependent responses associated with tumor TTF resistance. Tumor cell proliferation were assessed by labeling and tracking newly synthesized DNA during cell division (Fig. 5F–I). As shown in Fig. 5G, the tumor proliferation rate decreased with increasing field strength and reached a plateau after

~200 V/m, suggesting a the optimal TTF electric field strength for this patient was about 240 V/m. We also systematically studied tumor cell sensitivity to chemotherapeutic treatment (Temozolomide, TMZ and Anlotinib, AL) using the same microfluidic platform. Previous studies have explored the effects of TMZ dosage on glioblastoma, with dosages ranging from 100–1000 μM(Barciszewska, Gurda, Głodowicz, Nowak, and Naskręt-Barciszewska, 2015), while in clinical studies(Belter, Barciszewski, and Barciszewska, 2020), most patients were administered AL at concentrations ranging from 10–33 μM. We found that although TMZ exhibits a positive inhibitory effect on tumor cells, the cell proliferation rate remained around 38% with a strong inhibitory effect only at 800 μM, indicating that tumor cells are less sensitive to TMZ. On the contrary, AL exhibited a more pronounced inhibitory effect on tumor cells, with cell proliferation rates decreasing significantly with increasing concentration, almost reaching 0 when the concentration reached 60 μM. The optimal treatment parameter for the T-36 patient is 5 μM AL. These results provide a patient-based personalized treatment plan and provide guidance for better treatment of patients.

Due to the heterogeneity of tumors, patients may develop specific resistance to traditional therapies, and the same treatment regimen may be ineffective or overburdened for different patients, resulting in poor treatment effects or accelerated accumulation of tumor resistance. Thus, a patient-tailored patient response platform for brain tumor would facilitate therapy planning, provide patient-specific strategies and deliver optimal treatment. Our subsequent experimental explorations were tested with the best parameters specified by the patient, unless otherwise stated, the specific structured MIBTS with corresponding output will be designed and manufactured for subsequent processing.

### In vitro and in vivo validation

The inhibitory effect of MIBTS was validated with T-36 clinical patient cells. We comprehensively compared the inhibitory effects of MIBTS technology with Blank CTR, two different chemotherapy conditions, and TTF on clinical tumor cells (Fig. 6A). All groups involving drug therapy was set at the optimal therapeutic concentration (4 mM TMZ or 5 μM AL) screened for T-36 patient. As shown in Fig. 6B, MIBTS showed much enhanced therapeutic benefits compared to these traditional treatment techniques, with 80.1% (TMZ, $n = 3$, $p < 0.01$) and 72.2% (AL, $n = 3$, $p < 0.01$) higher suppression effect than first-line chemotherapy and 69.1% ($n = 3$, $p < 0.05$) higher than the most advanced tumor treating field technology.

The attenuation effect after implantation were tested in a porcine tissue (Fig. 6C), which is physiologically close to human tissue in terms of anatomy and composition. As shown in Fig. 6D, output attenuation increased slightly with the increase in implantation depth (13% attenuation at 10 mm to 23% attenuation at 50 mm) due to increased impedance in the tissue. The attenuation of the skull was calculated through simulation (Appendix Fig. S9), and indicated approximately 10–15% attenuation in mouse skulls and 39–53% attenuation in human skulls. We then validated the data in actual mouse skulls, which showed an attenuation of 12% on the mouse skull and is consistent with our simulated data. To further gauge the device's electrical attenuation with different implanted locations, we conducted tests at three distinct positions: atop the scalp, beneath the scalp but outside the

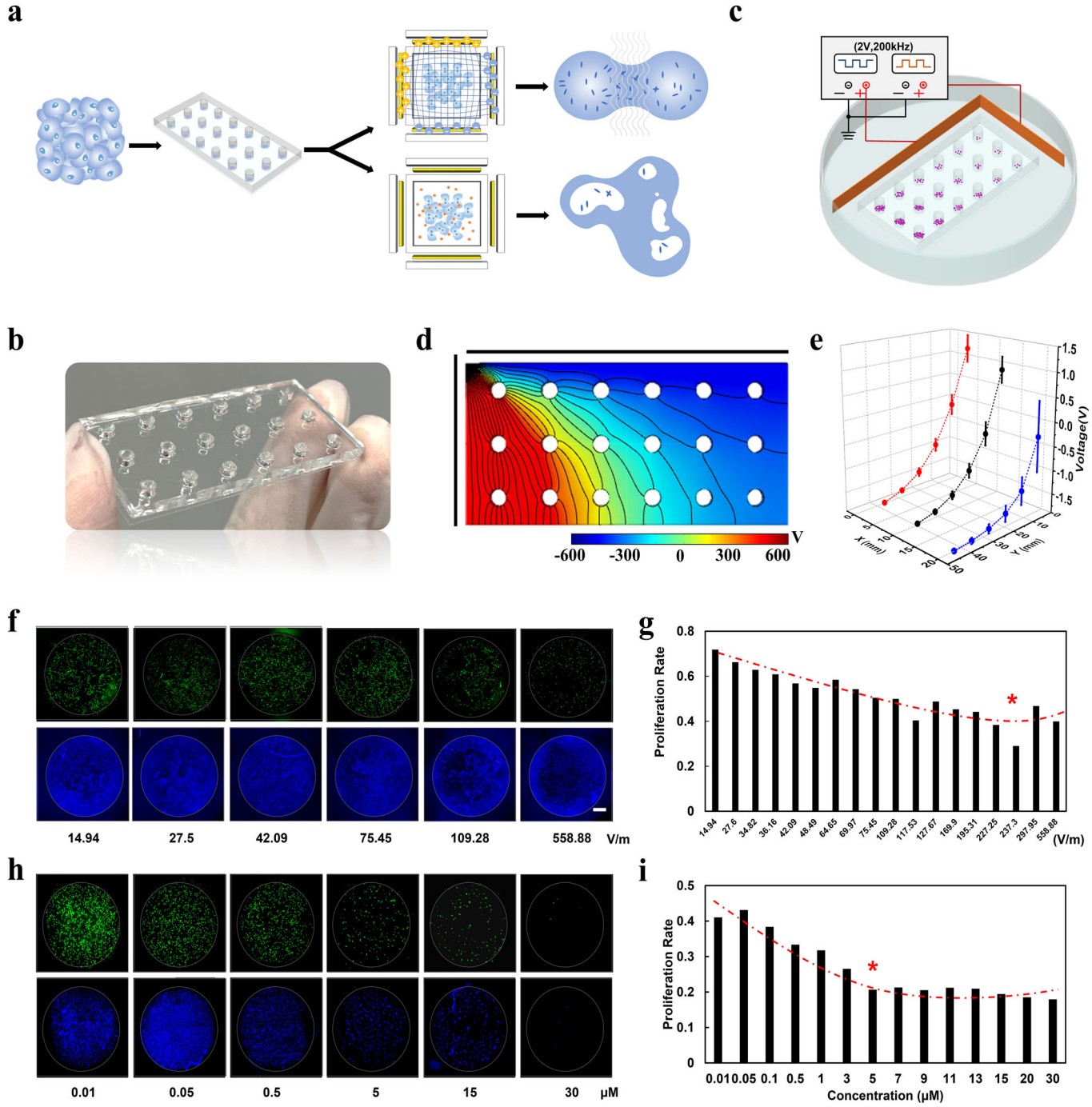

**Figure 5.  Patient-personalized MIBTS therapy.**

(A) A gradient microfluidic platform that incorporates arrays of cylindrical processing chambers to allow high-throughput exploration of TTF resistance and drug resistance in patients. (B) Photo of an 18-well microfluidic array platform. (C) The gradient microfluidic array platform (Fig. 5B) was connected to a homemade voltage regulation control unit. (D, E) Gradient voltage profiles generated in the microfluidic array platform. Error bars represent the SD from three devices. $n = 3$. (F) Fluorescence staining of newly synthesized DNA (green) and total DNA (blue) in all cells within samples under different TTF treatments. Scale bar: 500 μm. (G) TTF parameter optimization based on gradient voltage profiles. * indicates that the optimal TTF electric field strength for this patient is -240 V/m. (H) Fluorescence staining of newly synthesized DNA (green) and total DNA (blue) in all cells within samples under different drug treatments. Scale bar: 100 μm. (I) ISFCC parameter optimization based on gradient drug profiles. * indicates that the optimal treatment parameter is 5 μM AL. Source data are available online for this figure.

a

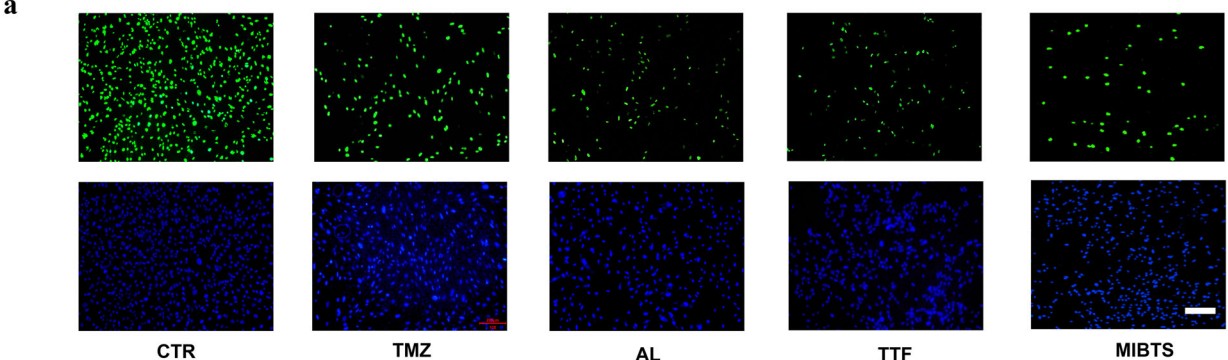

CTR          TMZ          AL          TTF          MIBTS

b

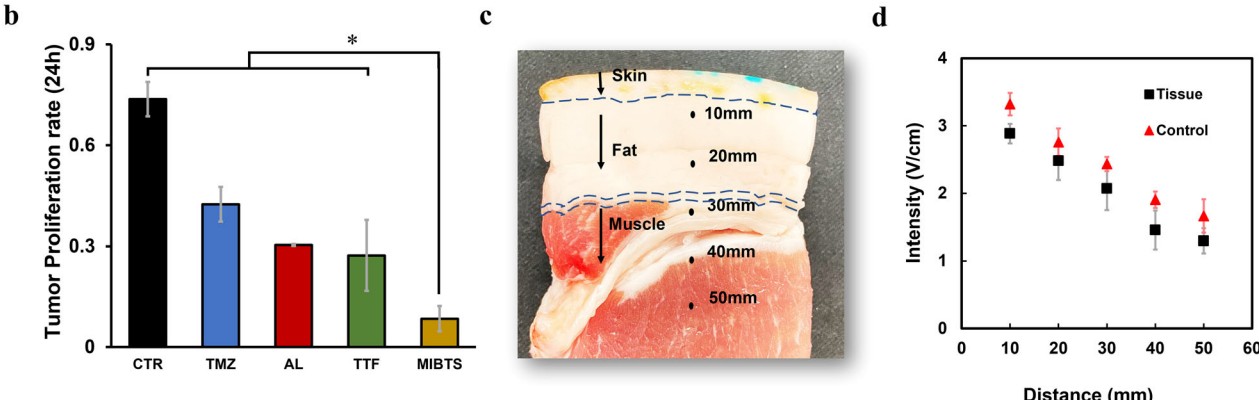

c

d

e

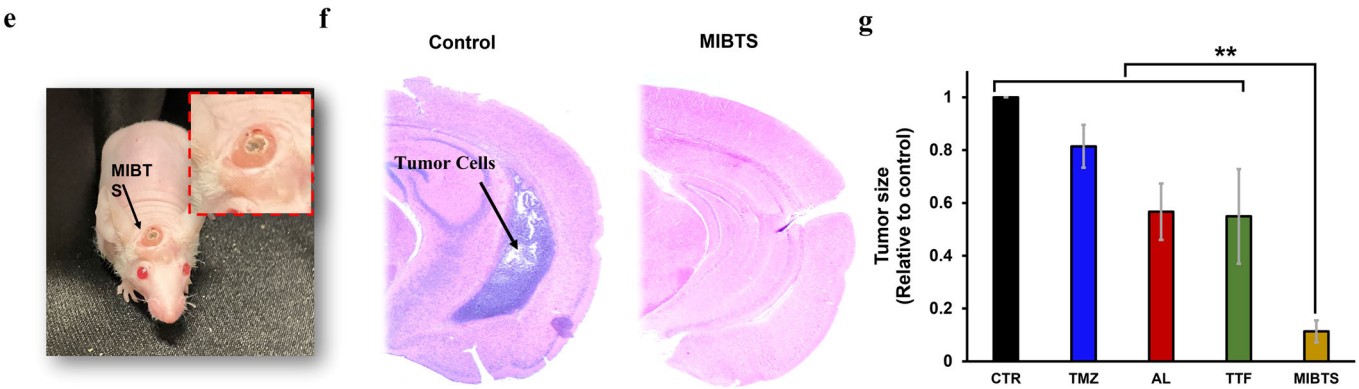

f

Control          MIBTS

g

**Figure 6.    In vitro and in vivo validation.**

(**A**) Fluorescence staining of newly synthesized DNA (green) and total DNA (blue) in all cells within samples under different therapy conditions. Scale bar: 200 µm. (**B**) MIBTS showed significantly enhanced therapeutic benefits compared to Blank CTR, chemotherapy drugs, and TTF. ***$p$ value = 0.002, 0.013, 0.017, 0.019, $t$-test, $n = 3$, error bars represent the standard deviation from three devices. (**C**) The attenuation effect of tissue tested in porcine tissue at different distances. (**D**) MIBTS output intensity at different distances with and without tissue. Error bars represent the SD from three devices. $n = 3$. (**E**) The MIBTS device is implanted through minimally invasive surgery with minimal trauma. (**F**, **G**) Histological studies showed a significant reduction in tumor area after MIBTS treatment. **$p$ value = 0.001, 0.013, 0.043, 0.048, $t$-test, $n = 3$, error bars represent the standard deviation from three devices. Scale bar: 0.5 mm. Source data are available online for this figure.

skull, and directly beneath the skull. As shown in Appendix Fig. S10, the scalp will cause about 4% of ultrasound attenuation, and the skull will cause about 12% of attenuation. Nevertheless, even when situated within tissue beneath the skull, the device persisted in generating an output signal of roughly 2.18 V/cm with an excitation power of 0.2 W/cm$^2$, ensuring an adequate TTF therapeutic treatment for the patients.

We also tested the therapeutic effect of MIBTS in tumor-bearing BALB/C-Nude mice with clinical patients' tumor in the brain. Experimental mice with similar tumor areas (coefficient of variation: 0.03) were randomly grouped, indicating a comparable tumor growth status between groups before any treatment. For therapy group, the MIBTS devices were implanted via minimally invasive surgery and delivery multimodal treatment for 21 days (Fig. 6E). It can be seen in Fig. 6F,G, all treatment groups showed a trend of tumor size reduction relative to the control group: the two treatment groups in the chemotherapy group was reduced by 18.5% ($n = 3$, $p > 0.05$) and 43.3% ($n = 3$, $p < 0.05$) respectively, TTF electric field treatment reduced the tumor area by 45.1% ($n = 3$, $p < 0.05$), and MIBTS treatment reduced the tumor area by 88.7% ($n = 3$, $p < 0.001$). Compared with the chemotherapy group, the treatment effects of the MIBTS group were improved by ~79.1% ($n = 3$, $p < 0.001$) and 51.2% ($n = 3$, $p < 0.01$), and MIBTS also showed an ~49.1% ($n = 3$, $p < 0.01$) improvement compared with the electric field treatment group. The above data not only demonstrate the successful positive therapeutic effect of the MIBTS treatment in vivo, but also verify the significantly superior of MIBTS to other treatment options. We also tracked tumor size for 14 days using in vivo fluorescence imaging, which confirmed that MIBTS therapy presented optimal therapeutic efficacy compared with other therapies (Appendix Fig. S11).

We explored the biological effects of ultrasound during MIBTS treatment, including its thermotoxic and beneficial therapeutic effects. It is reported that high-level exposure to ultrasound may cause thermal damage (Clarke and Haar, 1997) and resulting in tissue damages including: necrosis, apoptosis, and abnormal cell behavior (Ahmadi et al, 2012; Burks et al, 2011; Hinche et al, 2019b). A gentle and safe temperature increase were observed after hours of continuous ultrasound treatment (Appendix Fig. S12, the floating range was controlled within 0.17–1.36 °C at the power of 0.4 mW/cm$^2$), which ensured the safety of our therapy. The sham control group (tumor-bearing rats treated with ultrasound only) confirmed that no therapeutic or side effects were generated by the pure ultrasound treatment.

Further exploration was conducted into the long-term implantation safety and compatibility of the MIBTS. Daily observation and recording of the survival status of mice in each group were performed after establishing late-stage tumor models. Encouragingly, throughout the course of continuous treatment, mice in the MIBTS experimental group displayed a robust survival rate, no obvious toxicity has been induced to animals by the MIBTS implant, on the contrary, a significant increase in the survival time of mice with MIBTS implants were noticed (Appendix Fig. S13, $n = 5$, $p < 0.05$). Additionally, we tested and compared the pathological conditions of brain slices several months after implantation. No coagulation and inflammatory responses were induced (Appendix Figs. S14 and S15), confirming the excellent long-term biocompatibility of MIBTS implants.

## Conclusions

In this work, a promising minimally invasive intracranial brain tumor suppression (MIBTS) technology was presented for the effective treatment of brain cancer. Through minimally invasive implantation, this technology realizes the combined treatment of intracranial electric field and chemotherapy, which provides a new promising option for the treatment of brain tumors. Compared with existing technologies, this technology has shown great advantages in tumor suppressor effect, with 50–79% improvements compared with first-line chemo-treatment and 49% improvement compared to the most advanced membrane-based electrical field treating. Our MIBTS device enables precise volumetric control down to 1 cm$^3$, with therapeutic resolution thousands of times higher than conventional TTF(with a coarse spatial resolution of ~1450 cm$^3$ in human patients (Raven and Johnson, 2002)). In addition, the minimally invasive implantation (3 × 5 cm surgical window) ensures wider acceptance of the MIBTS devices. To achieve optimal patient-defined therapy and avoid the accumulation of tumor resistance, we packaged an 18-well microfluidic screening system in the platform to optimize therapy parameters. As an implanted device, the long-term biosafety of the MIBTS was investigated, which confirmed implant safety and good biocompatibility of the device. Meanwhile, the wireless operation allows the device to operate at a range of more than 50 mm subcutaneously with excitation ultrasound power about 0.2–0.4 W/cm$^2$, which is a safe level for most medical applications(Abrahao et al, 2019; Duck, 2007; Miller et al, 2012). The technique exhibited an over 80% inhibition of tumor growth and provides a new alternative to solve the current dilemma of brain tumor treatment. The technology is expected to change the treatment mode of brain tumors and provide new hope for controlling the recurrence of brain tumors with prolonged survival of patients. Here, we also need to admit that the current technology still needs more verification on humans in the future, such as the TTFields intensity and chemotherapy concentration in the gross brain tumor, and the therapeutic effect on the human body.

## Methods

**Reagents and tools table**

| Reagent/resource | Reference or source | Identifier or catalog number |
|---|---|---|
| **Experimental Models** | | |
| Anaplastic astrocytoma cells (*H. sapiens*) | The First Affiliated Hospital of Sun Yat-sen University | T-26 |
| Glioblastoma cells (*H. sapiens*) | The First Affiliated Hospital of Sun Yat-sen University | T-36 |
| Meningiomas cells (*H. sapiens*) | The First Affiliated Hospital of Sun Yat-sen University | T-51 |
| Glioblastoma cells (*H. sapiens*) | The First Affiliated Hospital of Sun Yat-sen University | T-59 |
| Oligodendroglioma cells (*H. sapiens*) | The First Affiliated Hospital of Sun Yat-sen University | T-91 |

| Reagent/resource | Reference or source | Identifier or catalog number |
|---|---|---|
| BALB/C-nu (*M. musculus*) | Sun Yat-sen University Animal Experiment Center | N/A |
| **Chemicals, enzymes, and other reagents** | | |
| Polydimethylsiloxane | Macklin | M934262 |
| Poly (DL-lactide-co-glycolide), Mw 24,000–38,000, 50:50 | Sigma-Aldrich | 739952-1 G |
| *N*-methyl-2-pyrrolidone | Macklin | N886189 |
| Temozolomide | The First Affiliated Hospital of Sun Yat-sen University | N/A |
| Anlotinib hydrochloride | The First Affiliated Hospital of Sun Yat-sen University | N/A |
| Phosphate-buffered saline | ScienCell | 0303 |
| Nutrient mixture F-12 | VivaCell | C3130-0500 |
| Fetal bovine serum | ScienCell | 0500 |
| L-glutamine | Aladdin | L432945 |
| Penicillin-streptomycin | MedChemExpress | HY-K1006 |
| Bovine serum albumin | Beyotime | ST2254 |
| Paraformaldehyde | Aladdin | C104190 |
| Triton-X | Meryer | M77656 |
| Hoechst | Aladdin | H412416 |
| EdU-click 488 | Sigma-Aldrich | BCK-EDU488 |
| Trypsin | Grand Island Biological | 11520626 |
| Isoflurane | KaiKule | N/A |
| Liquid paraffin | Macklin | P821317 |
| Ethanol | Aladdin | E111962 |
| Xylene | Aladdin | X112051 |
| Hematoxylin staining solution | Bioisco | HE011 |
| Differentiating solution | ScyTek | DSN999 |
| **Software** | | |
| COMSOL Multiphysics® 6.2 | https://cn.comsol.com/ | |
| Altium designer 24 | https://www.altium.com.cn/ | |
| **Other** | | |
| Hollow semi-rigid adhesive spacer | Keyuan | N/A |
| Flexible electrode layer | Keyuan | N/A |
| Spin coater | Huiyu | N/A |
| Oven | Ika | KW-4A |
| Temperature-controlled heating magnetic stirrer | As-One | IKA Oven 125 |
| Cell culture dish | Stemcell | 2-4994-01 |
| Cell incubator | iCell | 27150 |
| Biosafety cabinet | As-One | SG01-001 |
| Ultrasonic generato/ probe | Fuyida | FU-1500Q |
| Oscilloscope | Keysight | MSOX3104G |
| UV spectrophotometer | Sigma-Aldrich | C5416 |

| Reagent/resource | Reference or source | Identifier or catalog number |
|---|---|---|
| Scanning electron microscope | Aosive | Aosive 2020032661 |
| Plasma cleaner | Sumjune | SJVPR1001 |
| Water bath | JieMei | SYG-1210 |
| Centrifuge | DragonLab | LG-DL-82 |
| Small animal in vivo optical imaging system | Clinx | IVScope8200 |
| Microinjection pump | Hoyon | 20-0110 |
| Embedding machine | Beyotime | E0988 |
| Stereotaxic apparatus | Jkseiko | BJK-031 |
| Cranial drill | Accurate | Accu-GZ |
| Paraffin microtome | Beyotime | E0973 |
| Syringes of various sizes | LiGe | LG05 |
| Surgical instruments, mouse suturing needles and threads | JinHuan | 4-0# |

## Methods and protocols

### Clinical brain tumor cell culture

All study procedures related to clinical samples were approved by the Ethics Committee of Sun Yat-sen University and completed in strict accordance with relevant regulations (SYSU-IACUC-2020-B0157). Brain tumor tissues of patients (Appendix Table S1) were collected from the First Affiliated Hospital of SYSU (processed immediately after surgical resection), and informed consent was obtained from all patients. The experiments conformed to the principles set out in the WMA Declaration of Helsinki and the Department of Health and Human Services Belmont Report. Cell processing procedures were similar to our previous works. Clinical patients' cells were cultured in Dulbecco's Modified Eagle Medium/Nutrient Mixture F-12 (Hyclone) supplemented with fetal bovine serum (Hyclone), L-glutamine (Gibco), and penicillin-streptomycin solutions (Gibco). Cells were incubated at 37 °C in a humidified atmosphere with 5% $CO_2$, and the medium was changed every other day.

### Preparation of MIBTS

The cylindrical device is composed of seven layers of ultrathin films. A 50-μm flexible electrode layer was used as the internal cylindrical output layer, a PDMS film with a thickness of less than 50 μm was placed as the triboelectric pair layer with an adjustable gap controlled by a hollow half-rigid adhesive spacer from DEYI Company. Electrical connections and components are soldered to internal electrodes, and the entire device were coated with insulating layers and ISFI coatings. The integrated cylindrical device was about 8 mm long and less than 2 mm in diameter.

### Device characterization

The devices were placed in an external environment where a commercially available wireless ultrasonic energy converter (THD-M1) and generator (DK40) were used to adjust the ultrasonic power

density and the output frequency. The electrical outputs generated by the columnar device and customized TTF array were characterized using an oscilloscope (Rigol, DS2202) with a voltage probe (Rigol, RP3300, 1 MΩ input impedance). The ultrasonic power is determined by a calorimetric method, which measures the amount of heat transferred from the ultrasound.

## Simulations

We analyzed the ultrasonic, mechanical, and electrical conversion processes of the device using the finite element method in COMSOL. The physical parameters of PDMS, Au, air, water, and Cu were set to default values. After setting the environment parameters, a 5 cm circular ultrasound source were loaded on the top of the device with a distance from 10 to 50 mm. Three modules, including pressure acoustics, solid mechanics, and acoustic structural physics, are used to simulate the entire process and predict the acoustic and electrical outputs. The intracranial attenuation parameters were set according to the white matter attenuation coefficient: $\sigma = 0.15$ S/m; $\varepsilon = 3200$.

## Preparation of ISFI formulations

ISFI formulations are prepared with several controlled-release bioactive agents. Resomer® RG 503 H, Poly (DL-lactide-co-glycolide, Mw 24,000–38,000, 50:50, Sigma-Aldrich) was mixed with $N$-methyl-2- pyrrolidone (NMP, Aladdin) at a weight ratio of 1:10 (w/w), and allowed to dissolve by continuous mixing at room temperature for 24 h until a homogenous formulation is formed. Subsequently, chemotherapeutic reagents were added and stirred overnight at 37 °C to dissolve drugs and produce an in situ forming formulation with the desired drug concentration. According to the experience of previous in vitro experiments, we prepared three drug concentrations for the study. The final concentrations of TMZ in the ISFI formula were 4, 20, and 30 mM. The final concentrations of AL in the ISFI formula were 5, 20, and 30 μM.

## Fabrication of gradient microfluidic platform

Here we describe a microfluidic platform with well-controlled cylindrical chamber arrays to allow the construction of array-steep gradients for high-throughput chemical concentration assessment. We then constructed a high-throughput collection and feature extraction platform for electrophysiological response signals and Gradient compounds. A self-made TTF signal source based on CL8038 were used as source with the peripheral components adjusted (a reverse chip 74HC04D) for the frequency modulation and amplitude modulation of the signal. The PDMS solution was casted onto a previously 3D printed master plate with array of columns arranged uniformly, and then dried at 60 °C. We use a self-made electric field control module to realize gradient electric field regulation. Two sets of parallel plates are arranged on the periphery of the PDMS mold with external circuits connected to achieve a parallel tumor-treating electric field (TTF).

## In vitro experiment

In vitro validation experiments were performed in clinical tumor cells from patients treated with MIBTS for 24 h. The ultrasonic source is

### The paper explained

**Problem**

Current brain tumor treatment techniques face significant challenges due to the skull and blood–brain barrier (BBB), leading to a poor prognosis for patients with malignant brain tumors such as glioma. These issues result in a very short average survival time of less than two years for affected patients.

**Results**

We have developed an innovative minimally invasive brain tumor suppression (MIBTS) device that integrates patient-specific intracranial electric field therapy with an in-situ chemotherapeutic coating. This device is powered wirelessly by ultrasound, encapsulated in a chip-sized "Swiss-roll" structure, requires only a small surgical window ( ∼ 3 × 5 mm) for implantation. Our MIBTS device demonstrates up to 80% greater tumor suppression compared to first-line chemotherapy and 50–70% more efficacy than the latest tumor treating field technologies. Furthermore, it allows for personalized therapy adjustments without increasing the device size or complexity, ensuring optimal treatment and reducing the risk of tumor resistance. Long-term studies confirm the device's biosafety and biocompatibility, with safe operation at ultrasound power levels that are clinically acceptable.

**Impact**

The MIBTS device offers a groundbreaking approach to brain tumor treatment, addressing the limitations of current therapies with its minimally invasive design and superior efficacy. This technology not only improves the suppression of tumor growth but also holds promise for significantly extending patient survival times and reducing tumor recurrence. This technology has the potential to revolutionize brain tumor treatment and provide new hope for patients facing this challenging condition.

placed at a distance of 20 mm from the equipment, the ultrasonic power is 0.3 W/cm², and the electric field measured in the medium is 1.3–2 V/cm, 180 kHz. For parameter optimization, cells were seeded directly in microchambers and incubated with EdU (BeyoClick) for 24 h after establishing the electric field and drug gradient. The cells were washed, fixed, permeabilized, stained, and observed for cell proliferation under a fluorescent microscope (Leica, DiM8).

## In vivo experiment

All procedures involving animals were reviewed and approved by the Animal Ethics Committee of Sun Yat-Sen University. BALB/c-nude mice (20 ± 5 g) were purchased from Sun Yat-sen University Experiment Animal Center and randomly divided into four groups. Anesthetized mice were immobilized on a stereotaxic apparatus (RWD Life Science), and the skin above the head was incised. A hole with a diameter of less than 50 μm was drilled into the skull, and the prepared tumor cells were implanted into the left hemisphere of the mouse brain. The control group receives no treatment after tumor modeling. The chemotherapy group implants the chemotherapy drug AL, with a final concentration of 5 μM, near the tumor site through in situ injection. The TTF group implants the planar UP-TTD device externally on the skull. The device group implants the MIBTS device around the tumor and places the animal in a self-made ultrasonic emission chamber for treatment. After 27 days of treatment, the mice were anesthetized and perfused, and

brain tissue slices were taken out. Mouse brain sections were fixed, dehydrated, cleared, embedded in paraffin, and sectioned for physiological examination. About 10–15 coronary brain slice images were taken around the injection site, and the largest tumor area were selected for analysis.

## Data availability

This study includes no data deposited in external repositories.

The source data of this paper are collected in the following database record: biostudies:S-SCDT-10_1038-S44321-024-00091-5.

## Peer review information

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

## Acknowledgements

This work was supported by the National Natural Science Foundation of China (Grant number: 32101160), Foundation of Guangdong Provincial Key Laboratory of Sensor Technology and Biomedical Instrument (Grant number: 2020B1212060077).

## Author contributions

**Xiaona Cao**: Data curation; Formal analysis. **Jie Li**: Data curation; Validation. **Jinliang Ren**: Data curation; Formal analysis. **Jiajin Peng**: Data curation; Formal analysis. **Ruyue Zhong**: Data curation. **Jiahao He**: Data curation. **Ting Xu**: Data curation. **Zhenhua Yu**: Resources. **Huawei Jin**: Resources. **Siqi Hao**: Software. **Ruiwei Liu**: Software. **Bingzhe Xu**: Conceptualization; Supervision; Validation; Writing—original draft; Project administration; Writing—review and editing.

Source data underlying figure panels in this paper may have individual authorship assigned. Where available, figure panel/source data authorship is listed in the following database record: biostudies:S-SCDT-10_1038-S44321-024-00091-5.

