## [Peer Review File · EMBO Molecular Medicine]

Minimally-invasive Implantable Device Enhances Brain Cancer Suppression

Xiaona Cao, Jie Li, Jinliang Ren, Jiajin Peng, Ruyue Zhong, Jiahao He, Ting Xu, Zhenhua Yu, Huawei Jin, Siqi Hao, Ruiwei Liu, and Bingzhe Xu

Corresponding author(s): Bingzhe Xu (xubzh5@mail.sysu.edu.cn)

Review Timeline:

Submission Date:	20th Feb 24
Editorial Decision:	18th Mar 24
Revision Received:	25th Apr 24
Editorial Decision:	21st May 24
Revision Received:	3rd Jun 24
Accepted:	4th Jun 24

Editor: Zeljko Durdevic

Transaction Report:

18th Mar 2024

Dear Dr. Xu,

Thank you for the submission of your manuscript to EMBO Molecular Medicine. We have now received feedback from the two reviewers who agreed to evaluate your manuscript. As you will see from their reports pasted below, both referees acknowledge interest of the study but also raise serious concerns that preclude further consideration of the manuscript at this stage. Given the nature of referee criticisms addressing all the referees' comments would require a lot of additional work, time, and effort. Therefore, I am afraid that we do not feel it would be productive to call for a revised version of your manuscript.

Given the potential interest of the findings, we would, however, be willing to consider a new manuscript on the same topic if at some time in the near future you obtained data that would considerably strengthen the message of the study and address the referees concerns in full. To be completely clear, however, I would like to stress that if you were to send a new manuscript this would be treated as a new submission rather than a revision and would be reviewed afresh, in particular with respect to the literature and the novelty of your findings at the time of resubmission. If you decide to follow this route, please make sure you nevertheless upload a letter of response to the referees' comments.

I am sorry that I could not bring better news this time and hope that the referee comments are helpful in your continued work in this area.

Yours sincerely,

Zeljko Durdevic

***** Reviewer's comments *****

Referee #1 (Remarks for Author):

In this manuscript, the authors investigated a medical device for treating brain tumors that combines ultrasound-powered tumor treating field with intracranial chemotherapy release. They provide a technical characterization and some proof-of-principle in vitro and in vivo data.

This study addresses an unmet need for intracranial tumors of more effective therapies. It is a further development of a device that has been published before (Yang et al., An implantable ultrasound-powered device for treating brain cancer using electromagnetic fields. *Sci Adv.* 2022 Jul 22;8(29):eabm5023. doi: 10.1126/sciadv.abm5023. Epub 2022 Jul 22.).

Major

Figure 6A, B: How does the concentration of the TMZ control compare to the chemotherapy concentration in the MIBTS? This should be specified.

Figure 6F: was the control treated at all? Ideally, the in vivo experiment would have also been done with the 4 groups of the device with TTF (without ISFCC) or the device without UP-TFF but ISFCC and the combination thereof (MIBTS)?

Was there any toxicity in the in vivo experiments? A survival study would be ideal to judge appropriately about toxicity and also efficacy.

Minor

Figure 4d, how was the inhibition rate determined? In vitro or in vivo? The authors should add this.

Figure 6A, B, F. Which cell lines have been used? The authors should add this.

Referee #2 (Comments on Novelty/Model System for Author):

This is a novel device for the treatment of brain tumors. The novelty is based on the transduction of ultrasound to tumor treating fields, and there are a number of advantages. However, there are still technical issues and questions that need to be addressed. See my comments.

Referee #2 (Remarks for Author):

The authors designed and performed preclinical investigations on an implantable transducer that converts ultrasound energy into electric fields against tumors (aka tumor treating fields). This transducer array has a novel design as a Swiss-roll to increase the surface area that can capture the ultrasound energy for conversion into TTFIELDS. The authors also incorporated temozolomide into the biopolymer for simultaneous delivery in an effort to achieve increased tumor cell kill. The authors discussed the preclinical development of this device from fabrication to in vivo testing and concluded that it blocks glioblastoma growth by 99.9% in vitro and ~96% in vivo.

General Comments: TTFIELDS work by disrupting key biomolecules possessing a strong dipole moment that subserve a key cellular process. For example, TTFIELDS disrupt in a micromechanical fashion Septin and tubulin, which are key proteins that help coordinate and execute mitosis. Their disruption in tumor cells results in apoptosis or immunogenic cell death. Therefore, it seems silly to convert sound energy to electrical energy that ultimately disrupts cellular processes via biomechanical means. Still, this work is relevant in treating patients with glioblastoma since the source of ultrasound can be away from the skull and there is no direct contact from transducer arrays.

Another issue that I have with this work has to do with the enormous amount of data presented at a superficial level, which includes (i) MIBTS design, wireless tuning of TTFIELDS, (ii) coating of chemotherapy agents, (iii) patient-personalized MIBTS therapy (ex vivo platform to test for sensitivity of patient-specific tumor cells under TTFIELDS and chemotherapy agents), and (iv) validation of in vitro and in vivo models. Each one of these subtopics can potentially possess enough data for a full-length manuscript. This reviewer understands that the authors want to cure glioblastoma and other brain tumors, but having a deeper and more methodical evaluation of these components is still a worthwhile endeavor scientifically.

Specific Comments:

1. Abstract, First Paragraph of In-vitro and in-vivo validation, and Conclusion: Percentage should be expressed only up to the first decimal place. Expressing them to 2 decimal places, as in 112.12% and 42.82%, makes no biological sense.
2. Abstract: This statement is not correct, "... 99.9% inhibition of clinical glioblastoma growth in vitro ...". Take out the word clinical because in vitro finding is not a clinical finding. A clinical finding implies that it derives from patients or human subjects.
3. MIBTS design: This reviewer has 2 questions. First, how many turns in the Swiss-roll would be optimal to capture the maximum amount of ultrasound energy. I presume that increasing the number of rolls would cause the ultrasound to resonate within the Swiss-roll and this may generate an increase in electric fields. At the same time, heat is also generated and therefore what is the optimum design? Second, the authors only tested one chemotherapy agent, temozolomide. The biopolymer that holds temozolomide may have different affinity for other drugs, probably depending on their polarity and solubility. The authors have to characterize this aspect of the MIBTS design.
4. Wireless tuning of TTFIELDS: Skull is a major attenuator for TTFIELDS and ultrasound. Therefore, the authors have to do a sensitivity analysis using COMSOL to determine the magnitude of attenuation by increasing and decreasing the thickness of the skull in their nude mice first and then in humans.
5. Wireless tuning of TTFIELDS: Propagation of ultrasound waves follows the inverse squared law of physics. However, in Figure 3b, the extent of attenuation is linear rather than exponential by $1/r^2$. Please explain. May be the authors should do an experiment in which they implant the Swiss-roll (i) on top of the scalp, (ii) underneath the scalp but outside the skull, and (iii) just underneath the skull, and determine the extent of attenuation of ultrasound. This reviewer suspects that the reference setup (Swiss-roll just on top of the scalp) will follow exactly the inverse squared law but not the other 2 setups, which may have a greater attenuation due to scalp and bone absorption of ultrasound, unless the Swiss-roll configuration allows resonance to occur between membranes and therefore magnifies the ultrasound energy resulting in a higher electric field induction (this concept is similar to a step up electrical transformer). The authors have more experiments to do to answer this question.
6. Intracranial in situ forming chemotherapeutic coating: Is this coating dependent on the chemical characteristics of the chemotherapy drug, i.e. polar vs. non-polar drugs, solubility index?
7. Intracranial in situ forming chemotherapeutic coating: The authors did not show whether the release of drugs is from an ultrasound effect or by TTFIELDS. Please provide data.
8. Patient-personalized MIBTS therapy: The authors only tested one drug, temozolomide. They have to do at least 3 more with different mechanisms of actions, i.e. paclitaxel (tubulin inhibitor) and cisplatin (induction of dsDNA breaks), and bleomycin (a radiomimetic).

9. In-vitro and in-vivo validation: These data are not enough for human testing. One more step is needed, and that includes direct and COMSOL simulated measurements of TTFields intensity and chemotherapy concentration in the gross brain tumor, first in your nude mouse model and then in a few human subjects. Therefore, the authors have to state these limitations in Discussion.
10. Figure 3b: Please define the numbers (0, 20, 40, ..., 180, 200) on the x-axis of the graph.

As a service to authors, EMBO provides authors with the possibility to transfer a manuscript that one journal cannot offer to publish to another EMBO publication. The full manuscript and if applicable, reviewers reports are automatically sent to the receiving journal to allow for fast handling and a prompt decision on your manuscript. For more details of this service, and to transfer your manuscript to another EMBO title please click on Link Not Available

COMMENTS TO AUTHOR:

Referee: #1

*In this manuscript, the authors investigated a medical device for treating brain tumors that combines ultrasound-powered tumor treating field with intracranial chemotherapy release. They provide a technical characterization and some proof-of-principle in vitro and in vivo data. This study addresses an unmet need for intracranial tumors of more effective therapies. It is a further development of a device that has been published before (Yang et al., An implantable ultrasound-powered device for treating brain cancer using electromagnetic fields. *Sci Adv.* 2022 Jul 22;8(29):eabm5023. doi: 10.1126/sciadv.abm5023. Epub 2022 Jul 22.).*

We are very grateful for your efforts in reviewing our manuscript. Our point-by-point responses are detailed below.

1. *Figure 6A, B: How does the concentration of the TMZ control compare to the chemotherapy concentration in the MIBTS? This should be specified.*

We deeply appreciate the reviewer's comments. The TMZ concentration has been specified in the revised version on page 11, lines 6-8. All groups involving drug therapy was set at the optimal therapeutic concentration (4 mM TMZ) screened for T-36 patient.

2. *Figure 6F: was the control treated at all? Ideally, the in vivo experiment would have also been done with the 4 groups of the device with TTF (without ISFCC) or the device without UP-TFF but ISFCC and the combination thereof (MIBTS)?*

We thank the reviewer for the suggestions. The control group was not treated at all. As suggested by the reviewer, we have added more groups: TTF (without ISFCC) and the device without UP-TFF but ISFCC. In addition, we also have added new group with AL (without ISFCC) in the revised manuscript. It can be seen that all treatment groups showed a trend of tumor size reduction relative to the control group: the two treatment groups in the chemotherapy group were reduced by 18.5% (TMZ, n=3, p>0.05) and 43.3% (AL, n=3, p<0.05) respectively, TTF electric field treatment reduced the tumor area by 45.1% (n=3, p<0.05), and MIBTS treatment reduced the tumor area by 88.7% (n=3, p<0.001). Compared with the chemotherapy group, the treatment effects of the MIBTS group were improved by approximately 79.1% (n=3, p<0.001) and 51.2% (n=3, p<0.01), and MIBTS also showed an approximately 49.1% (n=3, p<0.01) improvement compared with the electric field treatment group. The above data not only demonstrate the success of MIBTS for tumor treatment, but also verify that this technology is significantly superior to other treatment options. We also tracked tumor size for 14 days using in vivo fluorescence imaging, which confirmed that MIBTS therapy presented the optimal therapeutic efficacy compared with other therapies (**Fig. S11**). Related discussion has been added on page 12, lines 8-20, page 40, lines 1-4.

Figure S11. *In-vivo* fluorescence imaging of tumor. (a) *In-vivo* fluorescence imaging of mice in each group. (b) Statistical results of average fluorescence intensity within 14 days, *** $p < 0.001$

3. Was there any toxicity in the *in vivo* experiments? A survival study would be ideal to judge appropriately about toxicity and also efficacy.

We thank the reviewer for the very professional suggestions. In the revised manuscript, we have conducted a survival study and added a further discussion on the long-term safety of MIBTS implant. In the survival study, no obvious toxicity has been induced to animals by the MIBTS implant, on the contrary, a significant increase in the survival time of mice with MIBTS implants were noticed (Fig. S13, $n=5$, $p < 0.05$). Additionally, we tested and compared the pathological conditions of brain slices one month after implantation. No coagulation and inflammatory responses were induced (Fig. S14 and Fig. S15), confirming the excellent long-term biocompatibility of MIBTS implants. Related discussion has been added on page 13, line 5-14, page 43-44.

Figure S13. Survival curves of four groups of mice with different therapy strategies.

Figure S14. No obvious coagulation and necrosis in the brain tissue were found after one-month MIBTS treatment. Scale bar 1 mm.

Figure S15. Long-term compatibility exploration of implanted MIBTS device after a month of implantation. No significant increase in leukomonocyte, macrophage, and polymorphonuclear leukocyte infiltration were observed in the MIBTS implanted animals. Scale bar 100 μm .

Minor

4. Figure 4d, how was the inhibition rate determined? *In vitro* or *in vivo*? The authors should add this.

Thanks for the comments. The inhibition rate was determined *in-vitro*. We have added related information on page 9, line 15.

5. Figure 6A, B, F. Which cell lines have been used? The authors should add this.

We appreciate the valuable suggestions. We have added the cell line information on Page 11, line 4. “The inhibitory effect of MIBTS was validated in T-36 patient cell lines.”

Referee: #2

This is a novel device for the treatment of brain tumors. The novelty is based on the transduction of ultrasound to tumor treating fields, and there are a number of advantages. However, there are still technical issues and questions that need to be addressed. See my comments.

The authors designed and performed preclinical investigations on an implantable transducer that converts ultrasound energy into electric fields against tumors (aka tumor treating fields). This transducer array has a novel design as a Swiss-roll to increase the surface area that can capture the ultrasound energy for conversion into TTFields. The authors also incorporated temozolomide into the biopolymer for simultaneous delivery in an effort to achieve increased tumor cell kill. The authors discussed the preclinical development of this device from fabrication to in vivo testing and concluded that it blocks glioblastoma growth by 99.9% in vitro and ~96% in vivo.

General Comments: TTFields work by disrupting key biomolecules possessing a strong dipole moment that subserve a key cellular process. For example, TTFields disrupt in a micromechanical fashion Septin and tubulin, which are key proteins that help coordinate and execute mitosis. Their disruption in tumor cells results in apoptosis or immunogenic cell death. Therefore, it seems silly to convert sound energy to electrical energy that ultimately disrupts cellular processes via biomechanical means. Still, this work is relevant in treating patients with glioblastoma since the source of ultrasound can be away from the skull and there is no direct contact from transducer arrays.

Another issue that I have with this work has to do with the enormous amount of data presented at a superficial level, which includes (i) MIBTS design, wireless tuning of TTFields, (ii) coating of chemotherapy agents, (iii) patient-personalized MIBTS therapy (ex vivo platform to test for sensitivity of patient-specific tumor cells under TTFields and chemotherapy agents), and (iv) validation of in vitro and in vivo models. Each one of these subtopics can potentially possess enough data for a full-length manuscript. This reviewer understands that the authors want to cure glioblastoma and other brain tumors, but having a deeper and more methodical evaluation of these components is still a worthwhile endeavor scientifically.

We appreciate the reviewer’s positive evaluation of our work and agree with the comments regarding the limitations of our study. As the reviewer mentioned, with acoustic-to-electrical conversion we can avoid direct contact of the transducer array from patients, which will enhance the safety of the technique. We also agree that some of our work needs to be further optimized, and we have done further scientifically endeavors with a deeper and more methodical evaluation of these components.

Specific Comments:

1. *Abstract, First Paragraph of In-vitro and in-vivo validation, and Conclusion: Percentage should be expressed only up to the first decimal place. Expressing them to 2 decimal places, as*

in 112.12% and 42.82%, makes no biological sense.

We are very grateful to the reviewer for the professional suggestions, and we have modified the decimal place of percentage data in the revised manuscript.

2. *Abstract: This statement is not correct, "... 99.9% inhibition of clinical glioblastoma growth in vitro ...". Take out the word clinical because in vitro finding is not a clinical finding. A clinical finding implies that it derives from patients or human subjects.*

Thanks for the comments. In the revised manuscript, we have deleted this description in the abstract.

3. *MIBTS design: This reviewer has 2 questions. First, how many turns in the Swiss-roll would be optimal to capture the maximum amount of ultrasound energy. I presume that increasing the number of rolls would cause the ultrasound to resonate within the Swiss-roll and this may generate an increase in electric fields. At the same time, heat is also generated and therefore what is the optimum design?*

Thanks for your very professional suggestion. In the revised version, we have added experiments and discussions on Swiss-roll optimization and performed further analysis on the heat generated by different number of rolls. We found that MIBTS with higher number of turns yield higher output voltages (Fig. S1a), which is due to the increased presence of interlayer interactions and multiple effects within the ultrasonic field. Temperature variations of devices with different turns have been also tested after continuous 8-hour ultrasonic stimulation (Fig. S1b). Although the number of turns does not significantly increase the heat of the device, excessive number of turns would definitely enlarge the bottom diameter of the device (Fig. S1c), which would complicate implant surgery and potentially increase patient trauma. Therefore, the appropriate number of turns should be tailored according to specific requirements for practical surgical operations. Related discussion has been added on page 5, lines 12-20, page 30, lines 1-4.

Figure S1. (a) Output voltages between Swiss-roll devices with different numbers of turns. (b) Temperature changes of devices with different loops after 8 hours of ultrasonic excitation. (c) The bottom diameters between Swiss-roll devices with different numbers of turns. *: p-value < 0.05; **: p-value < 0.01; ***: p-value < 0.001.

Second, the authors only tested one chemotherapy agent, temozolomide. The biopolymer that holds temozolomide may have different affinity for other drugs, probably depending on their polarity and solubility. The authors have to characterize this aspect of the MIBTS design.

Thanks for the comments. In the revised manuscript, we have added another chemotherapy agent AL3818 (Anlotinib Hydrochloride) for the drug release test (**Fig. S5**). Obviously, AL-coated ISFCs enters the stage of burst and continuous release earlier than their TMZ-coated counterparts. Numerous factors play a crucial role in the drug release process. It has been reported that burst release of drugs depends largely on in situ phase separation, which can be attributed to the physical and chemical properties of the drug, such as diffusion rate, distribution coefficient, dissociation constant, solubility, etc. [1,2]. In addition, it is also related to the hydrophobicity of the solvent, polymer type, molecular weight, etc. [3,4]. After the burst release phase, the polymer network gradually stabilizes and forms a more robust structure and drug molecules need to traverse a tighter polymer network to diffuse out. During this process, the drug diffuses and is released at a stable and sustained rate. At this stage, the main factors affecting drug release include drug diffusion rate within the polymer matrix, drug solubility, drug loading, matrix size (i.e., surface area of the solid implant), as well as its porosity and the tortuosity of diffusion pathways [1,5]. Related discussion has been added on page 8, lines 8-21, page 34, lines 1-2.

Figure S5. Drug release profiles of TMZ and AL over 24 hours

References:

- [1] Graham P D, Brodbeck K J, Mchugh A J. [J]. *Journal of Controlled Release*, 1999, 58(2): 233-245.
- [2] Higuchi W I. Analysis of Data on the Medicament Release from Ointments[J]. *Journal of Pharmaceutical Sciences*, 1962, 51(8): 802-804.
- [3] Ma D, Mchugh A J. The interplay of membrane formation and drug release in solution-cast films of polylactide polymers[J]. *International Journal of Pharmaceutics*, 2010, 388(1): 1-12.
- [4] Patel R B, Carlson A N, Solorio L, et al. [J]. *Journal of Biomedical Materials Research Part A*, 2010, 94(2): 476-84.
- [5] Faisant N, Siepmann J, Benoit J P. PLGA-based microparticles: elucidation of mechanisms and a new, simple mathematical model quantifying drug release[J]. *European Journal of Pharmaceutical Sciences*, 2002, 15(4): 355-366.

4. *Wireless tuning of TTFIELDS: Skull is a major attenuator for TTFIELDS and ultrasound. Therefore, the authors have to do a sensitivity analysis using COMSOL to determine the*

magnitude of attenuation by increasing and decreasing the thickness of the skull in their nude mice first and then in humans.

Thanks for the valuable comments, we very much agree that skull is a major attenuator in the clinical applications. In the revised manuscript, we have added a discussion of cranial attenuation on page 11, lines 17-20. The attenuation of the skull has been firstly analyzed using COMSOL (Fig. S9), and indicated approximately 10-15% attenuation in mouse skulls and 39-53% attenuation in human skulls. We then validated the data in actual mouse skulls, which showed an attenuation of ~12% on the mouse skull and is consistent with our simulated data.

Figure S9. Ultrasound attenuation under different skull thicknesses

5. *Wireless tuning of TTFields: Propagation of ultrasound waves follows the inverse squared law of physics. However, in Figure 3b, the extent of attenuation is linear rather than exponential by $1/r^2$. Please explain.*

May be the authors should do an experiment in which they implant the Swiss-roll (i) on top of the scalp, (ii) underneath the scalp but outside the skull, and (iii) just underneath the skull, and determine the extent of attenuation of ultrasound. This reviewer suspects that the reference setup (Swiss-roll just on top of the scalp) will follow exactly the inverse squared law but not the other 2 setups, which may have a greater attenuation due to scalp and bone absorption of ultrasound, unless the Swiss-roll configuration allows resonance to occur between membranes and therefore magnifies the ultrasound energy resulting in a higher electric field induction (this concept is similar to a step up electrical transformer). The authors have more experiments to do to answer this question.

Thanks for the professional suggestions. The reason for the failure to obey the inverse square law in Figure 3b is the small range of measured distances. We supplement the results of more extensive testing in Fig. S3. It can be seen that when the distance is expanded to 30 cm, attenuation still follows the inverse square law of physics. We have added related discussions on page 6, line 7-8, page 32, line 1.

Figure S3. Fitted curve of the attenuation of output voltage with increasing transmission distance.

As suggested by the reviewer, we have added output from different implant locations in the mouse to determine how much ultrasound attenuates in different tissues (page 11, lines 20-24, page 39, lines 1-3). As shown in **Fig. S10**, the scalp will cause about 4% of ultrasound attenuation, and the skull will cause about 12% of attenuation.

Figure S10. Device output at different implant locations

6. *Intracranial in situ forming chemotherapeutic coating: Is this coating dependent on the chemical characteristics of the chemotherapy drug, i.e. polar vs. non-polar drugs, solubility index?*

We thank the reviewer for the comments. The coating does not rely on the chemical properties of the chemotherapy drugs. When the ISFCC is implanted into the tumor site, the ISFCC components come into contact with the liquid in the tissue, triggering a series of phase change processes. The water-soluble solvent NMP will be rapidly absorbed or diluted by body fluids, causing the concentration of the biocompatible polymer PLGA to increase and precipitate, thereby forming a solid around the tumor. During the solid formation process, drugs will be released rapidly due to the instable polymer network, forming a burst release stage to ensure the drug takes effect quickly; Then over time, the polymer network gradually stabilizes to form a stronger structure, and drug molecules need to travel through a tighter polymer network to

diffuse out. During this process, the drug is diffusely released from the solid implant at a steady and sustained rate until the solid implant is completely degraded after several months. This process enables sustained-release drug delivery that lasts for several months, maintaining long-term effective treatment. Related discussion has been added on page 7, lines 18-25, page 8, lines 1-5.

7. *Intracranial in situ forming chemotherapeutic coating: The authors did not show whether the release of drugs is from an ultrasound effect or by TTF fields. Please provide data.*

Thank you for your comment. In the revised manuscript, we have added an exploration of the effect of ultrasound on drug release (page 8, lines 22-25, page 9, 1-8). Spontaneous and stable drug release could be identified as ISFCC coating degrades, therefore, the drug release from ISFCs is a spontaneous process that can occur even without excitation (**Fig. S6**). We also found that applying drugs onto coiled devices and subjecting them to ultrasound excitation would enhance their release rate.

Figure S6. Drug release profile of control and AL-coated MIBTS within 10 hours.

8. *Patient-personalized MIBTS therapy: The authors only tested one drug, temozolomide. They have to do at least 3 more with different mechanisms of actions, i.e. paclitaxel (tubulin inhibitor) and cisplatin (induction of dsDNA breaks), and bleomycin (a radiomimetic).*

We thank the reviewer for the valuable suggestions and have added more drugs to test (page 10, lines 8-19). Since the current first-line drug for the treatment of glioma is mainly TMZ, we only measured this drug before. In the revised version, we added testing of another drug, AL3818. Notably, the drug is still not a primary chemotherapy drug for glioma, but it has shown effectiveness in treating glioma in some studies. Previous studies have explored the effects of TMZ dosage on glioblastoma, with dosages ranging from 100~1000 μM , while in clinical studies, most patients were administered AL at concentrations ranging from 10~33 μM . We found that although TMZ exhibits positive inhibitory effect on tumor cells, the cell proliferation rate remained around 38% with strong inhibitory effect only at 800 μM , indicating that tumor cells are less sensitive to TMZ. On the contrary, AL exhibited a more pronounced inhibitory effect on tumor cells, with cell proliferation rates decreasing significantly with increasing concentration, almost reaching 0 when the concentration reached 60 μM . The optimal treatment parameter for the T-36 patient is 5 μM AL. These results provide a patient-based personalized

treatment plan and provide guidance for better treatment of patients.

9. *In-vitro and in-vivo validation: These data are not enough for human testing. One more step is needed, and that includes direct and COMSOL simulated measurements of TTFields intensity and chemotherapy concentration in the gross brain tumor, first in your nude mouse model and then in a few human subjects. Therefore, the authors have to state these limitations in Discussion.*

Thanks to the reviewers comments, we agree and have stated these limitations in the Discussion on page 14, lines 12-14. We also try to further discuss and augment changes in field strength in brain tissue (page 7, lines, 9-12). We also simulated the electric field distribution after MIBTS was implanted in the body, and its distribution of horizontal axis is shown in Figure S4, suggesting that the tumor-suppressing effect is lost after the electric field attenuates by about 4 cm.

Figure S4. Simulation of horizontal axis electric field distribution of MIBTS implanted in the brain

Figure 3b: Please define the numbers (0, 20, 40, ..., 180, 200) on the x-axis of the graph. Thank you for the valuable suggestions. The correction for Fig.3b has been made.

21st May 2024

Dear Dr. Xu,

Thank you for the submission of your revised manuscript to EMBO Molecular Medicine. I am pleased to inform you that we will be able to accept your manuscript pending the following final amendments:

1) In the main manuscript file, please do the following:

- Please address all comments suggested by our data editors listed below:

o Figure legends:

1. Please note that the legends for figures 5f, g, h, i are incorrectly labelled as 5e, f, g, h. This needs to be rectified.

2. Please define the annotated p values ***/**/* in the legend of figure 2e, g; 5g, i; 6b, g, as appropriate.

3. Please indicate the statistical test used for data analysis in the legends of figures 2e, g-h; 5g, i; 6b, g.

4. Please note that information related to n is missing in the legends of figures 2e, g-h; 3d; 4d-e; 5e; 6b, d, g.

5. Please note that the error bars are not defined in the legends of figures 2e, g-h; 2d; 4d-e; 5e; 6b, d, g.

6. Please note that the measure of center for the error bar needs to be defined in the legend of figure 3c.

7. Please note that the scale bar is missing for figure 5h.

8. Please note that the scale bar needs to be defined for figure 3i.

9. Please note that scale bar and its definition are missing for figures 6f.

- Remove all supplementary information.

- Rename "Competing interests" to "Disclosure and competing interests statement". We updated our journal's competing interests policy in January 2022 and request authors to consider both actual and perceived competing interests. Please review the policy <https://www.embopress.org/competing-interests> and update your competing interests if necessary.

- Author contributions: Please remove it from the manuscript and specify author contributions in our submission system. CRediT has replaced the traditional author contributions section because it offers a systematic machine-readable author contributions format that allows for more effective research assessment. You are encouraged to use the free text boxes beneath each contributing author's name to add specific details on the author's contribution. More information is available in our guide to authors:

<https://www.embopress.org/page/journal/17574684/authorguide#authorshippinguidelines>

- Please include structured Methods section that includes a Reagents and Tools Table followed by a Methods and Protocols section. File EV1 seems to be a detailed protocol in table format, please add it to the "Appendix" and rename tables to "Appendix Table S1" etc. and update the callouts in the text. Please check "Author Guidelines" for more information and to download table templates. <https://www.embopress.org/page/journal/17574684/authorguide#structuredmethods>

- Indicate in legends number and nature of replicates and exact p= values, not a range, along with the statistical test used. To keep the figures "clear" some authors found providing an Appendix table Sx with all exact p-values preferable. You are welcome to do this if you want to.

- In M&M, provide the statement that the informed consent was obtained from all human subjects and that the experiments conformed to the principles set out in the WMA Declaration of Helsinki and the Department of Health and Human Services Belmont Report.

- Correct the reference citation in the text and reference list. In the text a reference should be cited by author and year of publication. Include a space between a word and the opening parenthesis of the reference that follows. In the reference list, citations should be listed in alphabetical order. Where there are more than 10 authors on a paper, 10 will be listed, followed by "et al.". Please check "Author Guidelines" for more information.

<https://www.embopress.org/page/journal/17574684/authorguide#referencesformat>

- Please add data availability statement. If no data are deposited in public repositories, add the sentence: This study includes no data deposited in external repositories.

Please check "Author Guidelines" for more information.

<https://www.embopress.org/page/journal/17574684/authorguide#availabilityofpublishedmaterial>

2) Appendix: Please rename supplementary information to "Appendix", upload it as a single PDF file and add table of content with page numbers on the title page. The "Information of clinical cell lines" should be presented in a table format, named Appendix Table S1 and the table should be called out in the main manuscript text. Please also correct the nomenclature for the figures to "Appendix Figure S1" etc. and updated their callouts in the main manuscript text. Please delete the image of the movie file and the legend.

3) Movie: Please zip movie legend with the movie file and correct the nomenclature to "Movie EV1" and update the callouts in the text.

4) Funding: Please make sure that information about all sources of funding are complete in both our submission system and in the manuscript. Currently, project grant 2020B1212060077 is missing in our submission system.

5) The Paper Explained: Please provide "The Paper Explained" and add it to the main manuscript text. Please check "Author Guidelines" for more information. <https://www.embopress.org/page/journal/17574684/authorguide#researcharticleguide>

6) Synopsis: Every published paper now includes a 'Synopsis' to further enhance discoverability. Synopses are displayed on the journal webpage and are freely accessible to all readers. They include separate synopsis image and synopsis text.

- Synopsis image: Please provide a striking image or visual abstract as a high-resolution jpeg file 550 px-wide x (250-400)-px high to illustrate your article.
 - Synopsis text: Please provide a short standfirst (maximum of 300 characters, including space) as well as 2-5 one sentence bullet points that summarise the paper as a .doc file. Please write the bullet points to summarise the key NEW findings. They should be designed to be complementary to the abstract - i.e. not repeat the same text. We encourage inclusion of key acronyms and quantitative information (maximum of 30 words / bullet point). Please use the passive voice.
 - Please check your synopsis text and image before submission with your revised manuscript. Please be aware that in the proof stage minor corrections only are allowed (e.g., typos).
- 7) For more information: This space should be used to list relevant web links for further consultation by our readers. Could you identify some relevant ones and provide such information as well? Some examples are patient associations, relevant databases, OMIM/proteins/genes links, author's websites, etc...
- 8) As part of the EMBO Publications transparent editorial process initiative (see our Editorial at <http://embomolmed.embopress.org/content/2/9/329>), EMBO Molecular Medicine will publish online a Review Process File (RPF) to accompany accepted manuscripts. This file will be published in conjunction with your paper and will include the anonymous referee reports, your point-by-point response and all pertinent correspondence relating to the manuscript. Let us know whether you agree with the publication of the RPF and as here, if you want to remove or not any figures from it prior to publication. Please note that the Authors checklist will be published at the end of the RPF.
- 9) Please provide a point-by-point letter INCLUDING my comments as well as the reviewer's reports and your detailed responses (as Word file).

I look forward to reading a new revised version of your manuscript as soon as possible.

Yours sincerely,

Zeljko Durdevic

*** Instructions to submit your revised manuscript ***

- 1) a .docx formatted version of the manuscript text (including Figure legends and tables)
- 2) Separate figure files*
- 3) supplemental information as Expanded View and/or Appendix. Please carefully check the authors guidelines for formatting Expanded view and Appendix figures and tables at <https://www.embopress.org/page/journal/17574684/authorguide#expandedview>
- 4) a letter INCLUDING the reviewer's reports and your detailed responses to their comments (as Word file).
- 5) The paper explained: EMBO Molecular Medicine articles are accompanied by a summary of the articles to emphasize the

major findings in the paper and their medical implications for the non-specialist reader. Please provide a draft summary of your article highlighting

6) For more information: There is space at the end of each article to list relevant web links for further consultation by our readers. Could you identify some relevant ones and provide such information as well? Some examples are patient associations, relevant databases, OMIM/proteins/genes links, author's websites, etc...

7) Author contributions: the contribution of every author must be detailed in a separate section.

8) EMBO Molecular Medicine now requires a complete author checklist (<https://www.embopress.org/page/journal/17574684/authorguide>) to be submitted with all revised manuscripts. Please use the checklist as guideline for the sort of information we need WITHIN the manuscript. The checklist should only be filled with page numbers where the information can be found. This is particularly important for animal reporting, antibody dilutions (missing) and exact values and n that should be indicated instead of a range.

9) Every published paper now includes a 'Synopsis' to further enhance discoverability. Synopses are displayed on the journal webpage and are freely accessible to all readers. They include a short stand first (maximum of 300 characters, including space) as well as 2-5 one sentence bullet points that summarise the paper. Please write the bullet points to summarise the key NEW findings. They should be designed to be complementary to the abstract - i.e. not repeat the same text. We encourage inclusion of key acronyms and quantitative information (maximum of 30 words / bullet point). Please use the passive voice. Please attach these in a separate file or send them by email, we will incorporate them accordingly.

You are also welcome to suggest a striking image or visual abstract to illustrate your article. If you do please provide a jpeg file 550 px-wide x 300-800px high.

10) A Conflict of Interest statement should be provided in the main text

11) Please note that we now mandate that all corresponding authors list an ORCID digital identifier. This takes <90 seconds to complete. We encourage all authors to supply an ORCID identifier, which will be linked to their name for unambiguous name identification.

Currently, our records indicate that there is no ORCID associated with your account.

Please click the link below to provide an ORCID:

Link Not Available

Photos 400-800 DPI

*Additional important information regarding figures and illustrations can be found at <https://bit.ly/EMBOPressFigurePreparationGuideline>. See also figure legend preparation guidelines: <https://www.embopress.org/page/journal/17574684/authorguide#figureformat>

***** Reviewer's comments *****

Referee #1 (Remarks for Author):

The authors addressed each comment, performed additional experiments and adjusted the manuscript appropriately.

Referee #2 (Comments on Novelty/Model System for Author):

The authors have addressed my concerns adequately.

The authors have addressed all minor editorial requests.

4th Jun 2024

Dear Dr. Xu,

We are pleased to inform you that your manuscript is accepted for publication and is now being sent to our publisher to be included in the next available issue of EMBO Molecular Medicine.
